# THREATEN SPIKING NEURAL NETWORKS THROUGH COMBINING RATE AND TEMPORAL INFORMATION

**Zecheng Hao**[1], **Tong Bu**[1,2], **Xinyu Shi**[1,2], **Zihan Huang**[1], **Zhaofei Yu**[1,2]\* **& Tiejun Huang**[1,2]
[1] School of Computer Science, Peking University
[2] Institute for Artificial Intelligence, Peking University

## ABSTRACT

Spiking Neural Networks (SNNs) have received widespread attention in academic communities due to their superior spatio-temporal processing capabilities and energy-efficient characteristics. With further in-depth application in various fields, the vulnerability of SNNs under adversarial attack has become a focus of concern. In this paper, we draw inspiration from two mainstream learning algorithms of SNNs and observe that SNN models reserve both rate and temporal information. To better understand the capabilities of these two types of information, we conduct a quantitative analysis separately for each. In addition, we note that the retention degree of temporal information is related to the parameters and input settings of spiking neurons. Building on these insights, we propose a hybrid adversarial attack based on rate and temporal information (HART), which allows for dynamic adjustment of the rate and temporal attributes. Experimental results demonstrate that compared to previous works, HART attack can achieve significant superiority under different attack scenarios, data types, network architecture, time-steps, and model hyper-parameters. These findings call for further exploration into how both types of information can be effectively utilized to enhance the reliability of SNNs. Code is available at `https://github.com/hzc1208/HART_Attack`.

## 1 INTRODUCTION

Due to the unique event-driven property (Bohte et al., 2000) and superior biological plausibility (Gerstner et al., 2014), Spiking Neural Networks (SNNs) are acclaimed as the third generation of artificial neural networks (Maass, 1997) and have received extensive academic attention. Unlike traditional Analog Neural Networks (ANNs), SNNs utilize discrete binary sequences for communication between spiking neurons, with spikes being emitted only when the membrane potential exceeds the firing threshold. This characteristic makes SNNs well-suited for processing spatio-temporal data (Zhang & Li, 2020) and offers benefits such as reduced power consumption (Tavanaei et al., 2019; Zenke et al., 2021). In addition to image classification applications (Cao et al., 2015), SNNs have found utility in other areas such as text recognition (Lv et al., 2023) and object detection (Kim et al., 2020b), etc.

With the increasing deployment of SNNs on mobile devices and neuromorphic hardware (Davies et al., 2018; DeBole et al., 2019), ensuring the security and reliability of SNNs has become a critical concern. While SNNs exhibit stronger robustness compared to ANNs (Sharmin et al., 2020; Kundu et al., 2021; Ding et al., 2022), they are still susceptible to imperceptible attack samples crafted from learning gradient of SNNs, leading to potentially disastrous consequences, especially in safety-critical scenarios. Currently, both ANN-SNN conversion (Cao et al., 2015; Li et al., 2021) and Spatial-Temporal back-propagation (STBP) (Wu et al., 2018) can be leveraged for training SNNs, which respectively utilize the rate and temporal information of SNNs (Meng et al., 2022; Zhu et al., 2022). Here rate information mainly denotes the approximate linear transformation relationship between the average firing rate of adjacent layers, while temporal information refers to the information with time dimensions such as the order of spike sequence and the value of membrane potential at each time step. To achieve better attack performance, which method should be used to calculate the learning gradient? How should we utilize these two types of information reasonably?

---

\*Corresponding author: yuzf12@pku.edu.cn

Researchers have discovered that adopting rate or temporal information as auxiliary knowledge can further optimize the performance of SNNs (Kim et al., 2020a; Wang et al., 2022; Xu et al., 2023). In addition, theoretical validations have indicated that both rate and temporal information contribute to the overall characteristics of spike sequences (Panzeri & Schultz, 2001). The above findings hint that SNN models have the potential to retain and leverage both types of information, which is a significant difference between SNNs and ANNs. However, previous works (Sharmin et al., 2020; Ding et al., 2022; Bu et al., 2023) have not adequately designed gradient calculation methods to effectively integrate rate and temporal information into the adversarial attack framework of SNNs, thereby keeping the potential safety hazard of SNN models from being fully revealed.

In this paper, we present an in-depth analysis of rate and temporal information in SNNs. To better threaten SNNs, we propose a novel hybrid adversarial attack based on both rate and temporal information (HART). Experiments demonstrate the superiority of our attack method compared to previous strategies. To the best of our knowledge, this is the first work that simultaneously applies rate and temporal information to the adversarial attack of SNNs. Our main contributions are as follows:

- We redefine the rate information gradient, and the attack based on this gradient achieves better performance than the ANN-SNN Conversion attack. We quantitatively analyze the retention degree of temporal information in SNNs and identify its correlation with factors such as the membrane decay constant and the number of time-steps.
- We propose a hybrid adversarial attack based on both rate and temporal information (HART), which performs gradient pruning as well as merging on time dimension, offering an adjustable surrogate gradient and a pre-calculation property.
- We theoretically prove that our method has mathematical equivalence with the rate gradient. Additionally, we demonstrate the ability to control the temporal attribute of HART by adjusting the shape of the surrogate gradient curve.
- Extensive experiments validate the effectiveness of HART. Our method achieves state-of-the-art attack success rate (ASR) across various hyper-parameter settings for both static and neuromorphic datasets.

## 2   RELATED WORKS

**Learning algorithms for SNNs.** ANN-SNN conversion and STBP are currently the most widely adopted mainstream learning methods. Conversion methods are based on the principle of an approximately linear relationship between adjacent layers in SNNs, allowing researchers to train source ANNs and then replace their activation layers with spiking neurons to obtain SNNs (Cao et al., 2015; Rueckauer et al., 2017; Han et al., 2020). Presently, the converted SNNs can achieve comparable performance to ANNs on large-scale datasets with a sufficient number of time-steps (Deng & Gu, 2021). However, due to the presence of residual membrane potential, the mapping from ANNs to SNNs is not completely precise, resulting in performance degradation for converted SNNs under ultra-low time latency (Bu et al., 2022; Li et al., 2022; Hao et al., 2023a;b). Inspired by the back-propagation through time training mode in recurrent neural networks, researchers have proposed STBP for training SNNs (Shrestha & Orchard, 2018; Wu et al., 2018), which is a supervised learning algorithm that incorporates a time dimension. To tackle the non-differentiable issue during the spike firing procedure, surrogate gradient and various smoothing functions have been introduced (Neftci et al., 2019; Fang et al., 2021). In addition, hybrid training methods that leverage multiple types of information in SNNs have received widespread attention (Mostafa, 2017; Kim et al., 2020a; Zhang & Li, 2020; Wang et al., 2022). These hybrid approaches offer potential benefits in optimizing memory overhead and energy consumption in SNNs (Xiao et al., 2022; Rathi & Roy, 2023).

**Robustness and adversarial attack of SNNs.** Compared to ANNs, SNNs are considered to possess stronger robustness due to their capability to store diverse and rich information (Sharmin et al., 2020). Previous works aimed at improving the robustness of SNNs can be generally divided into two routes: one strengthens the model's robustness by migrating classic defense strategies from ANNs to SNNs, such as certification training (Liang et al., 2022) and Lipschitz analysis (Ding et al., 2022), while the other enhances defense capabilities by exploring some unique encoding and model specific to SNNs, including Poisson coding (Kundu et al., 2021; Leontev et al., 2021) and membrane time constant (El-Allami et al., 2021). Nevertheless, SNNs remain susceptible to adversarial attacks that exploit gradients learned during training. As both ANN-SNN conversion and STBP methods can be employed for model learning, Sharmin et al. (2019) explored the attack performance of the gradient

corresponding to these methods and found that the STBP attack is more effective. In addition, the STBP attack scheme adapted to neuromorphic data has also been designed (Lin et al., 2022; Marchisio et al., 2021). In contrast, Bu et al. (2023) proposed a method based on rate gradient approximation, achieving a higher attack success rate compared to STBP.

## 3 PRELIMINARIES

### 3.1 SPIKING NEURON MODELS

In this paper, we adopt the commonly used Integrate-and-Fire (IF) model and Leaky-Integrate-and-Fire (LIF) model (Gerstner & Kistler, 2002; Izhikevich, 2004). The dynamic equations about membrane potential in discrete form can be described as follows (Brette et al., 2007).

$$\boldsymbol{m}^l(t) = \lambda^l \boldsymbol{v}^l(t-1) + \boldsymbol{W}^l \boldsymbol{s}^{l-1}(t), \tag{1}$$

$$\boldsymbol{v}^l(t) = \boldsymbol{m}^l(t) - \boldsymbol{\eta}^l(t) \boldsymbol{s}^l(t), \tag{2}$$

$$\boldsymbol{\eta}^l(t) = \begin{cases} (\boldsymbol{m}^l(t) - v_{\text{rest}}), & \text{hard-reset} \\ \theta^l, & \text{soft-reset} \end{cases}, \tag{3}$$

$$\boldsymbol{s}^l(t) = \begin{cases} 1, & \boldsymbol{m}^l(t) \geqslant \theta^l \\ 0, & \text{otherwise} \end{cases}. \tag{4}$$

At the $t$-th time-step, we use the notation $\boldsymbol{m}^l(t)$ and $\boldsymbol{v}^l(t)$ to denote the membrane potential before and after triggering a spike, respectively. $\lambda^l$ is the membrane leaky constant. When $\lambda^l = 1$, the LIF model will degenerate into the IF model. $\boldsymbol{s}^l(t)$ determines whether to deliver a spike and $\boldsymbol{\eta}^l(t)$ indicates the type of reset for the neurons. Specifically, a hard-reset directly resets $\boldsymbol{v}^l(t)$ to the resting potential $v_{\text{rest}}$, while a soft-reset subtracts the corresponding threshold $\theta^l$ from $\boldsymbol{v}^l(t)$. $\boldsymbol{W}^l$ denote the weight matrix in the $l$-th layer, and $\boldsymbol{W}^l \boldsymbol{s}^{l-1}(t)$ represents the input current from layer $l-1$.

### 3.2 ADVERSARIAL ATTACK

Adversarial attack aims to exploit the vulnerability of neural networks by introducing maliciously crafted input data with imperceptible perturbations, causing the target model to make incorrect predictions (Goodfellow et al., 2015). Generally, it can be formulated as an optimization problem:

$$\arg\max_{\boldsymbol{\delta}} \mathcal{L}(f(\boldsymbol{x} + \boldsymbol{\delta}, \boldsymbol{W}), y), \text{ s.t. } ||\boldsymbol{\delta}||_p \leqslant \epsilon. \tag{5}$$

Here, $\boldsymbol{\delta}$ represents the adversarial perturbation, and its strength is constrained by $\epsilon$. $\mathcal{L}$ denotes the loss function based on the average firing rate. $f(\cdot)$ is the network model under attack, with internal weight parameters $\boldsymbol{W}$. $\boldsymbol{x}$ and $y$ denote the input images and the corresponding output target. In this paper, we consider two representative adversarial attack algorithms: the Fast Gradient Sign Method (FGSM) (Goodfellow et al., 2015) and Projected Gradient Descent (PGD) (Kurakin et al., 2017). White-box and black-box attacks, which respectively refer to the situation that hackers have access to the knowledge about model topology and parameters or not, are considered simultaneously.

The FGSM perturbs the input data linearly in the direction of the sign gradient with respect to the loss function, while PGD is a more powerful variant of FGSM, which enhances the attack effectiveness by optimizing the perturbation route iteratively. These methods can be described as follows:

$$\text{FGSM: } \hat{\boldsymbol{x}} = \boldsymbol{x} + \epsilon \, \text{sign}\left(\nabla_{\boldsymbol{x}} \mathcal{L}(f(\boldsymbol{x}, \boldsymbol{W}), y)\right), \tag{6}$$

$$\text{PGD: } \hat{\boldsymbol{x}}^k = \Pi_{\boldsymbol{x},\epsilon}\{\boldsymbol{x}^{k-1} + \alpha \, \text{sign}\left(\nabla_{\boldsymbol{x}} \mathcal{L}(f(\boldsymbol{x}^{k-1}, \boldsymbol{W}), y)\right)\}. \tag{7}$$

Here, $\epsilon$ limits the disturbance level of the input data, $k$ represents the iteration number, and $\alpha$ is the step size for each iteration. $\Pi_{\boldsymbol{x},\epsilon}$ denotes the constrained $\epsilon - l_p$ neighborhood projection space for $\boldsymbol{x}$.

### 3.3 RATE INFORMATION IN SPIKING NEURAL NETWORKS

Rate information in SNNs mainly refers to an approximate linear transformation relationship, similar to ANNs, between the average firing rate of adjacent layers. Here we use $\boldsymbol{r}^l(T) = \sum_{t=1}^{T} \boldsymbol{s}^l(t)/T$ to

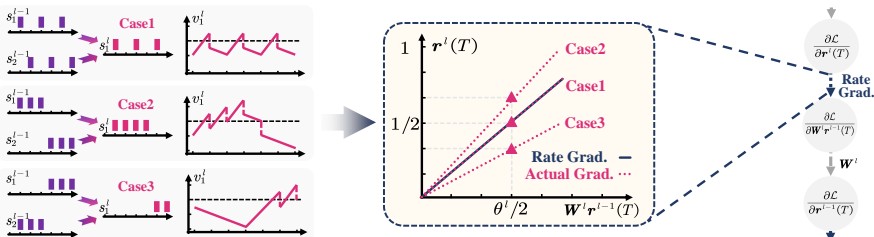

Figure 1: A set of examples for utilizing rate information in SNNs.

denote the average firing rate in layer $l$, with $T$ denoting the total number of time-steps. To simplify the expression, we adopt the soft-reset mechanism and set $\boldsymbol{v}^l(0) = 0, \theta^l = 1$ for each layer in SNNs. By combining Eqs. 1 and 2, summing from $t = 1$ to $t = T$, and dividing both sides by $T$, we get:

$$\boldsymbol{r}^l(T) = \boldsymbol{W}^l \boldsymbol{r}^{l-1}(T) - \frac{\boldsymbol{v}^l(T) + \sum_{t=1}^{T-1}(1 - \lambda^l)\boldsymbol{v}^l(t)}{T}. \tag{8}$$

Considering that $\boldsymbol{r}^l(T)$ is a non-negative vector, the above equation resembles the calculation rule $\boldsymbol{a}^l = \max(\boldsymbol{W}^l \boldsymbol{a}^{l-1}, 0)$ in ANNs with $\boldsymbol{a}^l$ denoting the activation output in layer $l$. At this point, the residual term $(\boldsymbol{v}^l(T) + \sum_{t=1}^{T-1}(1 - \lambda^l)\boldsymbol{v}^l(t))/T$ becomes the main gap from ANNs to SNNs. Specially, when considering the IF model ($\lambda^l = 1$), Eq. 8 simplifies to $\boldsymbol{r}^l(T) = \boldsymbol{W}^l \boldsymbol{r}^{l-1}(T) - \boldsymbol{v}^l(T)/T$, which forms the core idea of ANN-SNN conversion methods (Cao et al., 2015).

The specific value of $\boldsymbol{v}^l(t)(\forall t \in [1, T])$ is data-driven, resulting in varying residual terms for different neurons. When $\lambda^l = 1$, the residual term gradually approaches zero as the time-step increases, allowing for the training of high-precision SNNs in ANN-SNN conversion. However, this approach leads to performance degradation under ultra-low time latency conditions. For smaller values of $\lambda^l$, the residual term may not approach zero as $T$ increases, making its impact more significant. Therefore, from another perspective, we can consider the residual term as the temporal information retained by SNNs, as it is the primary characteristic that distinguishes SNNs from ANNs.

## 4 METHODS

In this section, we respectively analyze the influence of rate and temporal information on the robustness of SNN models. Then we propose a hybrid attack scheme named HART, which comprehensively leverages these two types of information. Additionally, we analyze some unique properties of HART related to gradient calculation.

### 4.1 RETHINKING RATE INFORMATION GRADIENT IN SNNS

Previous studies have demonstrated the potential of utilizing rate information for training non-leaky IF models and developing adversarial attack strategies that threaten the security of SNNs. However, the determination of the value of $\frac{\partial \boldsymbol{r}^l(T)}{\partial \boldsymbol{W}^l \boldsymbol{r}^{l-1}(T)}$ remains untackled, thus lacking an explicit and effective activation gradient for rate information. Considering that $\boldsymbol{a}^l = \max(\boldsymbol{W}^l \boldsymbol{a}^{l-1}, 0)$ and $\frac{\partial \boldsymbol{a}^l}{\partial \boldsymbol{W}^l \boldsymbol{a}^{l-1}}$ takes on 0 or 1 in ANNs, we aim to investigate the relationship between $\boldsymbol{r}^l(T)$ and $\boldsymbol{W}^l \boldsymbol{r}^{l-1}(T)$.

In contrast to ANNs, there is no deterministic relationship between $\boldsymbol{r}^l(T)$ and $\boldsymbol{W}^l \boldsymbol{r}^{l-1}(T)$. Fig. 1 illustrates this phenomenon using a simple set of examples, assuming a soft-reset mechanism and $\lambda^l = 1$. Despite receiving the same average input current $\boldsymbol{W}^l \boldsymbol{r}^{l-1}(T) = \theta^l/2$, Case 1-3 exhibit diverse average firing rates $\boldsymbol{r}^l(T)$, due to the different spike arrival sequence (more details are provided in the Appendix). According to Eq. 8, the actual gradient for the Rate-Input curve can be expressed as follows, visually represented by pink dashed lines in Fig. 1.

$$\frac{\boldsymbol{r}^l(T)}{\boldsymbol{W}^l \boldsymbol{r}^{l-1}(T)} = \boldsymbol{1} - \frac{\boldsymbol{v}^l(T) + \sum_{t=1}^{T-1}(1 - \lambda^l)\boldsymbol{v}^l(t)}{\boldsymbol{W}^l \sum_{t=1}^{T} \boldsymbol{s}^{l-1}(t)}. \tag{9}$$

Table 1: Attack success rate (ASR) of CBA and Ours (**bold font**) under white-box attack.

| Datasets | Time-steps | FGSM, $\lambda$=0.5 | FGSM, $\lambda$=1.0 | PGD, $\lambda$=0.5 | PGD, $\lambda$=1.0 |
|---|---|---|---|---|---|
| CIFAR-10 | 4 | 59.95/**86.42** | 64.95/**90.28** | 41.51/**99.08** | 52.65/**98.89** |
| | 8 | 60.40/**88.34** | 71.76/**92.56** | 42.13/**99.47** | 67.94/**99.90** |
| CIFAR10-DVS | 5 | 42.44/**49.92** | 37.39/**55.80** | 44.58/**55.57** | 42.46/**62.90** |
| | 10 | 36.05/**51.18** | 45.39/**74.47** | 38.95/**58.03** | 54.74/**89.61** |

Considering that $\frac{r^l(T)}{W^l r^{l-1}(T)}$ varies across different neurons, we propose Eq. 10 to relate the average firing rate in adjacent layers (solid blue line in Fig. 1), inspired by the principle of $\mathrm{ReLU}(\cdot)$.

$$g_{\text{rate}}^l = \left( \frac{\partial r^l(T)}{\partial W^l r^{l-1}(T)} \right)_{\text{rate}} = \begin{cases} \mathbb{E}\left( \frac{r^l(T)}{W^l r^{l-1}(T)} \right), & W^l \sum_{t=1}^{T} s^{l-1}(t) > 0 \\ 0, & \text{otherwise} \end{cases} . \quad (10)$$

Therefore, the gradient propagation chain based on rate information in SNNs can be described as:

$$(\nabla_{W^l} \mathcal{L})_{\text{rate}} = \frac{\partial \mathcal{L}}{\partial r^l(T)} g_{\text{rate}}^l r^{l-1}(T)^\top, \quad \left( \frac{\partial \mathcal{L}}{\partial r^{l-1}(T)} \right)_{\text{rate}} = \frac{\partial \mathcal{L}}{\partial r^l(T)} g_{\text{rate}}^l \frac{\partial W^l r^{l-1}(T)}{\partial r^{l-1}(T)}. \quad (11)$$

We use Conversion-based Approximation (CBA) (Sharmin et al., 2019) as a baseline, which is a conventional attack method relying on the approximation of the ANN-SNN conversion gradient. As shown in Tab. 1, we have demonstrated that our proposed activation gradient can outperform CBA in terms of attack performance. This result reinforces the importance of rate information in SNNs and validates the effectiveness of our proposed gradient.

### 4.2 ANALYZE THE RETENTION DEGREE OF TEMPORAL INFORMATION IN SNNS

As previously mentioned, the average firing rate $r^l(T)$ can exhibit a wide range of values even when the average input current $W^l r^{l-1}(T)$ is identical. Consequently, accurately characterizing the gradient of SNNs solely based on $\mathbb{E}\left( \frac{r^l(T)}{W^l r^{l-1}(T)} \right)$, which represents the mathematical expectation value derived from temporal information statistics, becomes challenging. Therefore, we attempt to measure the retention degree of temporal information in SNNs through the following equation.

$$\chi^l = \int_{-\infty}^{+\infty} \mathbf{Var}\left( \frac{r^l(T)}{W^l r^{l-1}(T)} \middle| W^l r^{l-1}(T) = x \right) \mathbf{P}\left( W^l r^{l-1}(T) = x \right) \mathrm{d}x. \quad (12)$$

Eq. 12 first divides spiking neurons into groups based on their average input current and then measures the richness of temporal information according to the expectation of the variance within each group. To facilitate subsequent theoretical analysis, we propose the following assumption.

**Assumption 1.** *Considering a group of spiking neurons with the same average input current $W^l r^{l-1}(T)$ and membrane leaky constant $\lambda^l$, we assume that for $\forall t \in [1, T]$, $v^l(t) \sim \mathbf{U}\left( g(W^l r^{l-1}(T), \lambda^l) - h(W^l r^{l-1}(T), \lambda^l), g(W^l r^{l-1}(T), \lambda^l) + h(W^l r^{l-1}(T), \lambda^l) \right)$. Here $g(\cdot)$ denotes the expectation of the uniform distribution, and $h(\cdot)$ represents the length of the distribution interval.*

Since Eq. 12 is not suitable for direct calculation and analysis, we present Theorem 1 based on Assumption 1. Detailed proof is provided in the Appendix.

**Theorem 1.** *If $W^l r^{l-1}(T) \sim \mathbf{U}(-c, c)$, for the soft-reset mechanism, we have $\chi^l = \int_{-c}^{c} \frac{[(T-1)(1-\lambda^l)^2+1]h^2(x,\lambda^l)}{6cT^2 x^2} \mathrm{d}x$. Moreover, assuming $h(x, \lambda^l) = ax + b$, we will further have $\chi^l = \frac{a^2 c^2 - b^2}{3c^2} \frac{(T-1)(1-\lambda^l)^2+1}{T^2}$.*

From Theorem 1, it can be found that the retention degree of temporal information is influenced by $\lambda$ or $T$ and improves with smaller values of $\lambda$ or $T$. To verify this conclusion, we conduct experiments on static and neuromorphic datasets (CIFAR-10 and CIFAR10-DVS) using different values of $\lambda$

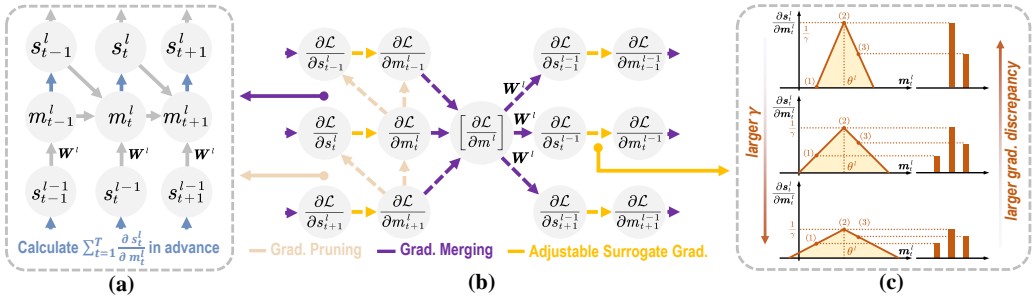

Figure 2: Overall algorithm framework for HART. (a): the property of pre-calculation, (b): back-propagation design, (c): adjustable temporal attribute.

and $T$. Since our attack scheme in Eqs. 10-11 is solely based on rate information, a higher attack success rate implies a more pronounced rate attribute of SNN models. Conversely, SNN models may exhibit more temporal attributes. As shown in Tab. 1, it can be observed that SNN models with (i) smaller $\lambda$, (ii) smaller $T$, and (iii) neuromorphic data exhibit better robustness and potentially retain more temporal information. These results indicate that SNNs still retain a certain degree of temporal characteristics, which motivates us to incorporate temporal components into SNN adversarial attacks.

### 4.3   Hybrid Adversarial Attack by Rate and Temporal Information (HART)

**Motivation.** Based on the preceding discussion, we have recognized the necessity of comprehensively utilizing both rate and temporal information. To effectively leverage these two types of information and enhance the performance of our attack, we propose the HART attack framework, as depicted in Fig. 2. In this framework, we maintain the standard forward-propagation pattern of spiking neurons, while modifying the back-propagation chain by pruning and merging it along the time dimension. This brand-new gradient calculation mode enables HART to capture rate information more effectively. In addition, we propose a surrogate function that can be adjusted based on the degree of temporal information retention in SNN models. This flexibility allows us to dynamically regulate the balance between rate and temporal attributes.

**Rate Attribute.** Our primary focus for pruning lies in $\partial \boldsymbol{m}^l(t+1)/\partial \boldsymbol{m}^l(t)$ and $\partial \boldsymbol{m}^l(t+1)/\partial \boldsymbol{s}^l(t)$, both of which play a crucial role in the gradient calculation process of STBP. However, if these two terms exist, $\partial \mathcal{L}/\partial \boldsymbol{m}^l(t)$ will receive contributions from $\partial \mathcal{L}/\partial \boldsymbol{m}^l(t+1), ..., \partial \mathcal{L}/\partial \boldsymbol{m}^l(T)$, which leads to strong correlations between gradients calculated at different time-steps, making it challenging to effectively capture rate information. To address this issue, as shown in Fig.2(b), we propose canceling $\partial \boldsymbol{m}^l(t+1)/\partial \boldsymbol{m}^l(t)$ and $\partial \boldsymbol{m}^l(t+1)/\partial \boldsymbol{s}^l(t)$. In addition, we further average $\partial \mathcal{L}/\partial \boldsymbol{m}^l(t), \forall t \in [1, T]$ before calculating $\partial \mathcal{L}/\partial \boldsymbol{s}^{l-1}(t)$ for all $t \in [1, T]$. By employing these operations, the gradient calculation chain of our proposed method can be expressed as follows.

$$\nabla_{\boldsymbol{W}^l}\mathcal{L} = \sum_{t=1}^{T}\left[\frac{\partial \mathcal{L}}{\partial \boldsymbol{m}^l}\right]\frac{\partial \boldsymbol{m}^l(t)}{\partial \boldsymbol{W}^l}, \quad \frac{\partial \mathcal{L}}{\partial \boldsymbol{s}^{l-1}(t)} = \left[\frac{\partial \mathcal{L}}{\partial \boldsymbol{m}^l}\right]\frac{\partial \boldsymbol{m}^l(t)}{\partial \boldsymbol{s}^{l-1}(t)}. \tag{13}$$

$$\left[\frac{\partial \mathcal{L}}{\partial \boldsymbol{m}^l}\right] = \frac{1}{T}\sum_{t=1}^{T}\frac{\partial \mathcal{L}}{\partial \boldsymbol{s}^l(t)}\frac{\partial \boldsymbol{s}^l(t)}{\partial \boldsymbol{m}^l(t)}. \tag{14}$$

At this point, we can demonstrate that the gradient calculation of our method in Eqs. 13-14 is equivalent to that based on rate information in Eq. 10 from the perspective of mathematical expectation. This equivalence verifies that our method possesses the attribute of effectively utilizing rate information, as stated in Theorem 2. A detailed proof is provided in the Appendix.

**Theorem 2.** *For spiking neurons with $\boldsymbol{W}^l\boldsymbol{r}^{l-1}(T) > 0$, when there exists an approximate proportional relationship between $g(\boldsymbol{W}^l\boldsymbol{r}^{l-1}(T), \lambda^l)$ and $\boldsymbol{W}^l\boldsymbol{r}^{l-1}(T)$ (denoted as $k^l = \frac{g(\boldsymbol{W}^l\boldsymbol{r}^{l-1}(T), \lambda^l)}{\boldsymbol{W}^l\boldsymbol{r}^{l-1}(T)}$), under soft-reset mechanism, if $\mathbb{E}\left(\sum_{t=1}^{T}\frac{\partial \boldsymbol{s}^l(t)}{\partial \boldsymbol{m}^l(t)}\right) = T - k^l((T-1)(1-\lambda^l)+1)$, then we have $\mathbb{E}\left(\nabla_{\boldsymbol{W}^l}\mathcal{L}\right) = (\nabla_{\boldsymbol{W}^l}\mathcal{L})_{rate}$ and $\mathbb{E}\left(\sum_{t=1}^{T}\frac{\partial \mathcal{L}}{\partial \boldsymbol{s}^{l-1}(t)}\right) = \left(\frac{\partial \mathcal{L}}{\partial \boldsymbol{r}^{l-1}(T)}\right)_{rate}$. Here, $\mathbb{E}\left(\nabla_{\boldsymbol{W}^l}\mathcal{L}\right)$ and*

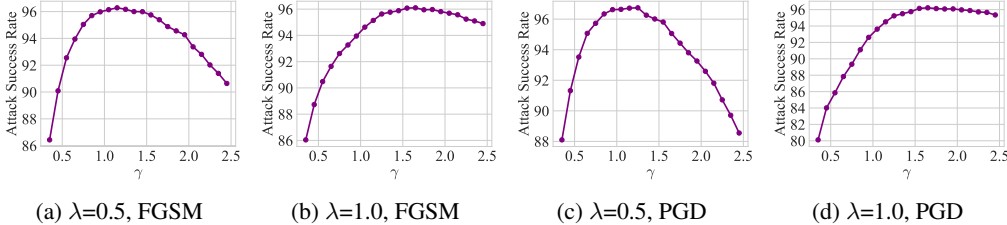

| (a) $\lambda$=0.5, FGSM | (b) $\lambda$=1.0, FGSM | (c) $\lambda$=0.5, PGD | (d) $\lambda$=1.0, PGD |

Figure 3: The performance of HART under different $\gamma$ on CIFAR-10.

Table 2: Comparison between HART and previous works under white-box attack (WBA). * denotes robust target models.

| Dataset | Architecture | $\lambda$ | Clean Acc. | Attack | CBA | BPTR | STBP | RGA | Ours |
|---|---|---|---|---|---|---|---|---|---|
| CIFAR-10 | VGG-11 | 0.5 | 91.48 | FGSM | 60.40 | 82.67 | 91.71 | 93.63 | **96.28** |
| | | | | PGD | 42.13 | 99.21 | 99.95 | 99.92 | **100.00** |
| | | 0.9 | 93.03 | FGSM | 70.58 | 88.36 | 89.91 | 94.41 | **97.24** |
| | | | | PGD | 55.29 | 99.45 | 99.94 | 99.97 | **99.98** |
| | | 0.9* | 89.99 | FGSM | 25.49 | 41.77 | 55.41 | 56.76 | **58.70** |
| | | | | PGD | 20.77 | 61.45 | 78.55 | 74.42 | **83.54** |
| | | 1.0 | 93.06 | FGSM | 71.76 | 88.76 | 86.28 | 93.74 | **96.22** |
| | | | | PGD | 67.94 | 99.63 | 99.70 | 99.94 | **99.97** |
| | ResNet-17 | 0.9 | 93.04 | FGSM | 44.29 | 85.06 | 84.24 | 92.93 | **94.80** |
| | | | | PGD | 29.76 | 99.86 | 99.91 | 100.00 | **100.00** |
| CIFAR-100 | VGG-11 | 0.9 | 73.28 | FGSM | 83.73 | 92.47 | 92.88 | 94.72 | **96.06** |
| | | | | PGD | 82.91 | 99.59 | 99.86 | 99.92 | **99.96** |
| | | 0.9* | 67.21 | FGSM | 32.69 | 57.19 | 70.42 | 70.24 | **72.41** |
| | | | | PGD | 27.57 | 71.98 | 86.56 | 83.35 | **87.68** |
| | ResNet-17 | 0.9 | 72.05 | FGSM | 65.34 | 86.94 | 85.66 | 92.06 | **94.54** |
| | | | | PGD | 45.17 | 99.65 | 99.69 | 99.90 | **99.96** |
| CIFAR10-DVS | VGG-DVS | 0.5 | 76.0 | FGSM | 36.05 | 50.39 | 59.08 | 53.95 | **61.05** |
| | | | | PGD | 38.95 | 60.00 | 71.05 | 62.11 | **74.08** |
| | | 1.0 | 76.0 | FGSM | 45.39 | 69.74 | 76.97 | 76.05 | **78.42** |
| | | | | PGD | 54.74 | 87.11 | 92.63 | 89.08 | **93.03** |

$\mathbb{E}\left(\sum_{t=1}^{T} \frac{\partial \mathcal{L}}{\partial \boldsymbol{s}^{l-1}(t)}\right)$ *refer to the gradient obtained after calculating the mean value of the surrogate gradient term in the back-propagation chain.*

**Temporal Attribute.** Similar to Wu et al. (2018), we address the non-differentiable issue arising from $\partial \boldsymbol{s}^l(t)/\partial \boldsymbol{m}^l(t)$ by using a surrogate gradient function, as illustrated in Eq. 15. In this equation, the parameter $\gamma$ controls the shape of the surrogate function.

$$\frac{\partial \boldsymbol{s}^l(t)}{\partial \boldsymbol{m}^l(t)} = \frac{1}{\gamma^2} \max\left(\gamma - |\boldsymbol{m}^l(t) - \theta^l|, 0\right). \tag{15}$$

By combining Assumption 1 and Eqs.13-15, we can adjust the expected gradient $\mathbb{E}\left(\sum_{t=1}^{T} \frac{\partial \mathcal{L}}{\partial \boldsymbol{s}^{l-1}(t)}\right)$ for a group of neurons with the same average input current by modifying the value of $\gamma$. From Eq. 15, it can be derived that the slope of the surrogate function is $1/\gamma^2$, and the corresponding membrane potential range is $[\theta^l - \gamma, \theta^l + \gamma]$. Moreover, we observe that the position of $\sum_{t=1}^{T} \frac{1}{\gamma^2} \max\left(\gamma - |\boldsymbol{m}^l(t) - \theta^l|, 0\right)$ in the back-propagation chain is equivalent to the position of $\boldsymbol{g}_{\text{rate}}^l$ in Eq. 11 or $\frac{\partial \boldsymbol{a}^l}{\partial \boldsymbol{W}^l \boldsymbol{a}^{l-1}}$ in ANNs. Therefore, we can consider it as the activation gradient of HART. Consequently, as illustrated in Fig. 2(c), a larger value of $\gamma$ results in a smoother surrogate gradient curve, a wider membrane potential range covered, and a smaller discrepancy in the activation gradient obtained by groups of neurons with different average input currents or spike arrival sequences. In this scenario, our method can be considered to have a gradient with more rate attributes. Conversely, when $\gamma$ becomes smaller, HART focuses on the subset of spiking neurons belonging to a specific membrane potential range in each time-step, allowing us to obtain a gradient with more temporal attributes.

**Pre-calculation Property.** It is important to note that in the HART framework, the condition $\forall l, \frac{\partial \mathcal{L}}{\partial \boldsymbol{s}^l(1)} = ... = \frac{\partial \mathcal{L}}{\partial \boldsymbol{s}^l(T)}$ is actually satisfied. Therefore, according to Eqs.13-14, we can consider

Table 3: Comparison between HART and previous works under black-box attack (BBA). * denotes robust target models.

| Dataset | Architecture | $\lambda$ | Clean Acc. | Attack | CBA | BPTR | STBP | RGA | Ours |
|---|---|---|---|---|---|---|---|---|---|
| CIFAR-10 | VGG-11 | 0.5 | 91.48 | FGSM | 43.04 | 63.44 | 77.77 | 79.65 | **82.68** |
| | | | | PGD | 23.50 | 84.21 | 95.99 | 95.36 | **96.74** |
| | | 0.9 | 93.03 | FGSM | 43.45 | 66.72 | 73.45 | 77.28 | **85.82** |
| | | | | PGD | 23.98 | 84.72 | 95.04 | 94.69 | **97.62** |
| | | 0.9* | 89.99 | FGSM | 14.08 | 25.26 | 35.83 | 35.44 | **38.26** |
| | | | | PGD | 10.63 | 31.10 | 46.06 | 44.42 | **47.83** |
| | | 1.0 | 93.06 | FGSM | 43.28 | 64.25 | 68.03 | 73.26 | **80.34** |
| | | | | PGD | 24.75 | 80.55 | 90.91 | 91.36 | **96.22** |
| | ResNet-17 | 0.9 | 93.04 | FGSM | 36.07 | 69.53 | 67.11 | 80.11 | **84.95** |
| | | | | PGD | 15.57 | 93.72 | 94.30 | 98.36 | **99.28** |
| CIFAR-100 | VGG-11 | 0.9 | 73.28 | FGSM | 68.33 | 80.10 | 80.90 | 84.27 | **88.51** |
| | | | | PGD | 42.45 | 88.91 | 93.65 | 93.91 | **97.32** |
| | | 0.9* | 67.21 | FGSM | 22.59 | 37.58 | 47.20 | 47.94 | **50.78** |
| | | | | PGD | 18.24 | 41.73 | 54.40 | 54.78 | **57.66** |
| | ResNet-17 | 0.9 | 72.05 | FGSM | 61.22 | 75.65 | 74.30 | 81.19 | **85.31** |
| | | | | PGD | 32.59 | 91.07 | 89.13 | 95.66 | **98.06** |
| CIFAR10-DVS | VGG-DVS | 0.5 | 76.0 | FGSM | 34.87 | 44.08 | 47.89 | 48.55 | **49.74** |
| | | | | PGD | 35.13 | 47.63 | 50.53 | 50.92 | **53.16** |
| | | 1.0 | 76.0 | FGSM | 43.03 | 62.50 | 66.32 | 65.79 | **69.74** |
| | | | | PGD | 52.11 | 70.92 | 76.45 | 75.66 | **78.03** |

pre-calculating $\sum_{t=1}^{T} \frac{\partial \boldsymbol{s}^l(t)}{\partial \boldsymbol{m}^l(t)}$ during the forward propagation stage, allowing us to update the gradients in a single operation during the back-propagation. By leveraging this property, we can reduce the cost of calculating gradients for HART from $O(T)$ to $O(1)$.

**Empirical Principles for Selecting $\gamma$.** We have previously discovered that when using a smaller value of $\lambda$, SNNs tend to exhibit more temporal characteristics. Therefore, in order to achieve better attack effectiveness, it is advisable to use a smaller value of $\gamma$ at this time. This viewpoint is supported by Fig. 3, where it can be seen that CIFAR-10/$\lambda = 1.0$ has a larger $\gamma$ while CIFAR-10/$\lambda = 0.5$ has a smaller $\gamma$ under the condition of achieving optimal attack success rate. Additionally, from Fig. 3, we observe that the curve of ASR with respect to $\gamma$ approximately follows an unimodal distribution. Based on the aforementioned empirical principles, we can approximately determine the optimal value of $\gamma$ by selecting several points with equal intervals within the search range.

## 5 EXPERIMENTS

### 5.1 EXPERIMENTAL SETUP

We validate the effectiveness of our proposed attack method on the CIFAR-10/100 (Krizhevsky et al., 2009) and CIFAR10-DVS (Li et al., 2017) datasets. We attempt different values of $\lambda$, time-steps, perturbation degree and network architectures (Simonyan & Zisserman, 2014; Zheng et al., 2021). In our experiments, we employed FGSM and PGD as the basic attack methods in both white-box and black-box environments. Previous studies (Sharmin et al., 2019; 2020) have pointed out that SNN models trained through STBP exhibit stronger robustness compared to ANN-SNN conversion, such as (Ding et al., 2022). Therefore, we select a set of SNN models as our attack targets, trained using the STBP method for 8 time-steps (CIFAR-10, CIFAR-100) or 10 time-steps (CIFAR10-DVS). In addition, we also consider the effect of robust training, high-intensity attacks, filter, and encoding scheme on our attack framework. More detailed experimental configuration is provided in Appendix.

### 5.2 COMPARISON WITH PREVIOUS STATE-OF-THE-ART WORKS

To demonstrate the superiority of our proposed method, we compare it with the current state-of-the-art techniques, including CBA (Sharmin et al., 2019), STBP (Sharmin et al., 2020), BPTR (Ding et al., 2022), and RGA (Bu et al., 2023). We utilize the attack success rate as a metric to evaluate the effectiveness of these different attacks. As shown in Tabs. 2 and 3, our method consistently achieves optimal performance across all experimental scenarios, whether under a white-box attack or black-box attack. This demonstrates the robustness and generalization capability of our approach, considering the different attack environments, data types, network architectures, and $\lambda$ values. Notably, our

Table 4: ASR for STBP/RGA/HART with different time-steps on CIFAR-10/VGG-11.

| $\lambda$ | Time-steps | FGSM, WBA | FGSM, BBA | PGD, WBA | PGD, BBA |
|---|---|---|---|---|---|
| | 4 | 90.07/93.24/**95.68** | 76.22/78.52/**80.10** | 99.88/99.85/**99.98** | 94.59/94.21/**94.96** |
| 0.5 | 8 | 91.71/93.63/**96.28** | 77.77/79.65/**82.68** | 99.92/99.92/**100.00** | 95.99/95.36/**96.74** |
| | 16 | 91.86/93.48/**95.82** | 77.49/79.66/**83.49** | 99.95/99.91/**99.99** | 96.12/95.98/**97.29** |
| | 4 | 81.89/91.03/**92.67** | 65.52/71.24/**76.43** | 99.17/99.23/**99.40** | 87.48/89.37/**92.71** |
| 1.0 | 8 | 86.28/93.74/**96.22** | 68.03/73.26/**80.34** | 99.70/99.94/**99.97** | 90.91/91.36/**96.22** |
| | 16 | 87.49/95.24/**96.65** | 66.89/75.07/**81.41** | 99.88/99.97/**99.99** | 90.86/92.67/**97.14** |

Table 5: ASR for STBP/RGA/HART with different perturbation degrees on CIFAR-10/VGG-11.

| $\lambda$ | $\epsilon$ | FGSM, WBA | FGSM, BBA | PGD, WBA | PGD, BBA |
|---|---|---|---|---|---|
| | 2/255 | 49.15/45.76/**55.91** | 24.67/22.87/**26.41** | 66.32/62.08/**78.33** | 29.30/28.42/**30.50** |
| 0.5 | 4/255 | 76.30/76.86/**83.06** | 51.28/50.05/**54.31** | 96.99/95.14/**98.95** | 69.43/68.12/**71.54** |
| | 8/255 | 91.71/93.63/**96.28** | 77.77/79.65/**82.68** | 99.92/99.92/**100.00** | 95.99/95.36/**96.74** |
| | 2/255 | 46.41/44.46/**46.76** | 19.19/19.62/**21.89** | **65.58**/61.44/65.26 | 21.89/21.96/**24.75** |
| 1.0 | 4/255 | 71.82/75.17/**78.56** | 41.48/42.76/**47.80** | 95.28/95.27/**96.39** | 57.29/56.78/**64.08** |
| | 8/255 | 86.28/93.74/**96.22** | 68.03/73.26/**80.34** | 99.70/99.94/**99.97** | 90.91/91.36/**96.22** |

method exhibits particular advantages in black-box attack scenarios. For instance, when $\lambda = 0.9$ and employing the FGSM attack, our method outperforms RGA by 2.83% under the white-box attack setting and surpasses RGA by 8.54% under the black-box attack setting. Moreover, since our models are trained using the STBP method, the gradient corresponding to the STBP attack can be considered the most accurate gradient estimation. However, our method outperforms the STBP attack, indicating that SNN models preserve both rate and temporal information. Therefore, directly applying the principles of adversarial attacks in ANNs cannot achieve optimal results for SNNs.

## 5.3 Performance of HART with Different Time-steps

To account for potential time constraints or the necessity of selecting different time-steps in practical attack scenarios, we conduct additional experiments to evaluate the performance of our method across a range of time-steps, from 4 to 16. As mentioned earlier, the retention degree of temporal information in SNN models may vary at different time-steps. However, we can dynamically adjust the temporal attribute of the HART attack by tuning the parameter $\gamma$. Tab. 4 presents the results of these experiments. Our method consistently outperforms both the STBP and RGA techniques in all experimental cases, showcasing its general adaptive capability.

## 5.4 Performance of HART under Different Perturbation Degree

Our proposed method also exhibits remarkable superiority in adversarial attacks with limited perturbations, as demonstrated by the results presented in Tab. 5. For example, when $\lambda = 0.5$, $\epsilon = 2/255$, and adopting a white-box attack, the performance gap between HART and previous techniques is 6.76% for the FGSM attack and 12.01% for the PGD attack. These results highlights the precision of the gradient direction obtained from HART and underscores its ability to pose a significant threat to SNNs even within an extremely small step size range.

## 6 Conclusions

In this paper, we first conduct a quantitative analysis of rate and temporal information in SNNs, respectively, and then propose an adversarial attack method that integrates these two types of information. We emphasize the simultaneous preservation of both rate and temporal information in SNNs, and the considerable advantages of our proposed method validate this standpoint. These findings motivate further research on effectively integrating rate and temporal information into the adversarial training of SNNs, which is an interesting research direction includes drawing inspiration from the characteristics of the brain's neural circuits to achieve robustness similar to that of the human brain (Yu et al., 2020; 2024). In addition, the potential application of these gradient techniques to SNN training is also a fascinating direction.

ACKNOWLEDGEMENTS

This work was supported by the National Natural Science Foundation of China (62176003, 62088102) and by Beijing Nova Program (20230484362).

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

## A   APPENDIX

### A.1   PROOF OF THEOREM

**Theorem 1.** *If $W^l r^{l-1}(T) \sim U(-c, c)$, for soft-reset mechanism, we have $\chi^l = \int_{-c}^{c} \frac{[(T-1)(1-\lambda^l)^2+1]h^2(x,\lambda^l)}{6cT^2x^2}\mathrm{d}x$. Moreover, assuming $h(x,\lambda^l) = ax + b$, we will further have $\chi^l = \frac{a^2c^2-b^2}{3c^2}\frac{(T-1)(1-\lambda^l)^2+1}{T^2}$.*

*Proof.* Based on Assumption 1 and the preconditions in Theorem 1, by combining Eq. 8 and Eq. 12, we will have

$$
\begin{aligned}
\chi^l &= \int_{-\infty}^{+\infty} \mathbf{Var}\left(1 - \frac{v^l(T) + \sum_{t=1}^{T-1}(1-\lambda^l)v^l(t)}{W^l\sum_{t=1}^{T}s^{l-1}(t)}\right)\mathbf{P}\left(W^l r^{l-1}(T) = x\right)\mathrm{d}x \\
&= \int_{-c}^{c} \frac{\mathbf{Var}\left(v^l(T)\right) + \sum_{t=1}^{T-1}(1-\lambda^l)^2\mathbf{Var}\left(v^l(t)\right)}{2cT^2x^2}\mathrm{d}x \\
&= \int_{-c}^{c} \frac{[(T-1)(1-\lambda^l)^2+1]h^2(x,\lambda^l)}{6cT^2x^2}\mathrm{d}x.
\end{aligned}
\tag{A1}
$$

With the assumption that $h(x, \lambda^l) = ax + b$, we will derive the following equation.

$$
\begin{aligned}
\chi^l &= \int_{-c}^{c} \frac{[(T-1)(1-\lambda^l)^2+1]h^2(x,\lambda^l)}{6cT^2x^2}\mathrm{d}x \\
&= \frac{(T-1)(1-\lambda^l)^2+1}{6cT^2} \int_{-c}^{c}\left(a^2 + \frac{2ab}{x} + \frac{b^2}{x^2}\right)\mathrm{d}x \\
&= \frac{a^2c^2-b^2}{3c^2}\frac{(T-1)(1-\lambda^l)^2+1}{T^2}.
\end{aligned}
\tag{A2}
$$

$\square$

**Theorem 2.** *For spiking neurons with $W^l r^{l-1}(T) > 0$, when there exists an approximate proportional relationship between $g(W^l r^{l-1}(T), \lambda^l)$ and $W^l r^{l-1}(T)$ (denoted as $k^l = \frac{g(W^l r^{l-1}(T), \lambda^l)}{W^l r^{l-1}(T)}$), under soft-reset mechanism, if $\mathbb{E}\left(\sum_{t=1}^{T}\frac{\partial s^l(t)}{\partial m^l(t)}\right) = T - k^l((T-1)(1-\lambda^l)+1)$, then we have $\mathbb{E}\left(\nabla_{W^l}\mathcal{L}\right) = \left(\nabla_{W^l}\mathcal{L}\right)_{rate}$ and $\mathbb{E}\left(\sum_{t=1}^{T}\frac{\partial\mathcal{L}}{\partial s^{l-1}(t)}\right) = \left(\frac{\partial\mathcal{L}}{\partial r^{l-1}(T)}\right)_{rate}$. Here, $\mathbb{E}\left(\nabla_{W^l}\mathcal{L}\right)$ and $\mathbb{E}\left(\sum_{t=1}^{T}\frac{\partial\mathcal{L}}{\partial s^{l-1}(t)}\right)$ refer to the gradient obtained after calculating the mean value of the surrogate gradient term in the back-propagation chain.*

*Proof.* Assuming an SNN with $L$ layers, we first consider the case of the output layer ($l = L$). We can derive that $\sum_{t=1}^{T}\frac{\partial\mathcal{L}}{\partial s^L(t)} = \sum_{t=1}^{T}\frac{\partial\mathcal{L}}{\partial r^L(T)}\frac{\partial r^L(T)}{\partial s^L(t)} = \frac{1}{T}\sum_{t=1}^{T}\frac{\partial\mathcal{L}}{\partial r^L(T)} = \frac{\partial\mathcal{L}}{\partial r^L(T)}$ and $\frac{\partial\mathcal{L}}{\partial s^L(1)} = \frac{\partial\mathcal{L}}{\partial s^L(2)} = ... = \frac{\partial\mathcal{L}}{\partial s^L(T)}$. With the preconditions in Theorem 2, we get the following equation

according to Eqs. 13-15.

$$
\begin{aligned}
\mathbb{E}\left(\nabla_{\boldsymbol{W}^L}\mathcal{L}\right) =& \mathbb{E}\left(\left(\frac{1}{T}\sum_{t=1}^{T}\frac{\partial\mathcal{L}}{\partial\boldsymbol{s}^L(t)}\frac{\partial\boldsymbol{s}^L(t)}{\partial\boldsymbol{m}^L(t)}\right)\left(\sum_{t=1}^{T}\frac{\partial\boldsymbol{m}^L(t)}{\partial\boldsymbol{W}^L}\right)\right) \\
=& \mathbb{E}\left(\frac{1}{T}\frac{\partial\mathcal{L}}{\partial\boldsymbol{r}^L(T)}\sum_{t=1}^{T}\frac{\partial\boldsymbol{s}^L(t)}{\partial\boldsymbol{m}^L(t)}\boldsymbol{r}^{L-1}(T)^{\top}\right) \\
=& \frac{1}{T}\frac{\partial\mathcal{L}}{\partial\boldsymbol{r}^L(T)}\mathbb{E}\left(\sum_{t=1}^{T}\frac{\partial\boldsymbol{s}^L(t)}{\partial\boldsymbol{m}^L(t)}\right)\boldsymbol{r}^{L-1}(T)^{\top} \\
=& \frac{\partial\mathcal{L}}{\partial\boldsymbol{r}^L(T)}\frac{[T-k^L((T-1)(1-\lambda^L)+1)]}{T}\boldsymbol{r}^{L-1}(T)^{\top} \\
=& \frac{\partial\mathcal{L}}{\partial\boldsymbol{r}^L(T)}\mathbb{E}\left(1-\frac{\boldsymbol{v}^L(T)+\sum_{t=1}^{T-1}(1-\lambda^L)\boldsymbol{v}^L(t)}{\boldsymbol{W}^L\sum_{t=1}^{T}\boldsymbol{s}^{L-1}(t)}\right)\boldsymbol{r}^{L-1}(T)^{\top} \\
=& \frac{\partial\mathcal{L}}{\partial\boldsymbol{r}^L(T)}\left(\frac{\partial\boldsymbol{r}^L(T)}{\partial\boldsymbol{W}^L\boldsymbol{r}^{L-1}(T)}\right)_{\text{rate}}\boldsymbol{r}^{L-1}(T)^{\top} \\
=& \left(\nabla_{\boldsymbol{W}^L}\mathcal{L}\right)_{\text{rate}}.
\end{aligned}
\tag{A3}
$$

By following a similar derivation process as in Eq. A3, we can obtain $\mathbb{E}\left(\sum_{t=1}^{T}\frac{\partial\mathcal{L}}{\partial\boldsymbol{s}^{L-1}(t)}\right) = \left(\frac{\partial\mathcal{L}}{\partial\boldsymbol{r}^{L-1}(T)}\right)_{\text{rate}}$. In the HART method, due to the presence of merge operations, for any layer in SNNs, we have $\frac{\partial\mathcal{L}}{\partial\boldsymbol{s}^l(t)} = \left[\frac{\partial\mathcal{L}}{\partial\boldsymbol{m}^{l+1}}\right]\frac{\partial\boldsymbol{m}^{l+1}(t)}{\partial\boldsymbol{s}^l(t)}$, where $t = 1,...,T$. Consequently, we can derive that $\frac{\partial\mathcal{L}}{\partial\boldsymbol{s}^l(1)} = \frac{\partial\mathcal{L}}{\partial\boldsymbol{s}^l(2)} = ... = \frac{\partial\mathcal{L}}{\partial\boldsymbol{s}^l(T)}$. Based on the above conclusions, for any layer in SNNs, if $\mathbb{E}\left(\sum_{t=1}^{T}\frac{\partial\mathcal{L}}{\partial\boldsymbol{s}^l(t)}\right) = \left(\frac{\partial\mathcal{L}}{\partial\boldsymbol{r}^l(T)}\right)_{\text{rate}}$ is already satisfied and each term in the gradient calculation chain is independent of each other, we can further derive the following equation.

$$
\begin{aligned}
\mathbb{E}\left(\nabla_{\boldsymbol{W}^l}\mathcal{L}\right) =& \mathbb{E}\left(\left(\frac{1}{T}\sum_{t=1}^{T}\frac{\partial\mathcal{L}}{\partial\boldsymbol{s}^l(t)}\frac{\partial\boldsymbol{s}^l(t)}{\partial\boldsymbol{m}^l(t)}\right)\left(\sum_{t=1}^{T}\frac{\partial\boldsymbol{m}^l(t)}{\partial\boldsymbol{W}^l}\right)\right) \\
=& \frac{1}{T}\mathbb{E}\left(\sum_{t=1}^{T}\frac{\partial\mathcal{L}}{\partial\boldsymbol{s}^l(t)}\right)\mathbb{E}\left(\sum_{t=1}^{T}\frac{\partial\boldsymbol{s}^l(t)}{\partial\boldsymbol{m}^l(t)}\right)\boldsymbol{r}^{l-1}(T)^{\top} \\
=& \frac{\partial\mathcal{L}}{\partial\boldsymbol{r}^l(T)}\left(\frac{\partial\boldsymbol{r}^l(T)}{\partial\boldsymbol{W}^l\boldsymbol{r}^{l-1}(T)}\right)_{\text{rate}}\boldsymbol{r}^{l-1}(T)^{\top} \\
=& \left(\nabla_{\boldsymbol{W}^l}\mathcal{L}\right)_{\text{rate}},
\end{aligned}
\tag{A4}
$$

$$
\begin{aligned}
\mathbb{E}\left(\sum_{t=1}^{T}\frac{\partial\mathcal{L}}{\partial\boldsymbol{s}^{l-1}(t)}\right) =& \mathbb{E}\left(\left(\frac{1}{T}\sum_{t=1}^{T}\frac{\partial\mathcal{L}}{\partial\boldsymbol{s}^l(t)}\frac{\partial\boldsymbol{s}^l(t)}{\partial\boldsymbol{m}^l(t)}\right)\left(\sum_{t=1}^{T}\frac{\partial\boldsymbol{m}^l(t)}{\partial\boldsymbol{s}^{l-1}(t)}\right)\right) \\
=& \frac{1}{T}\mathbb{E}\left(\sum_{t=1}^{T}\frac{\partial\mathcal{L}}{\partial\boldsymbol{s}^l(t)}\right)\mathbb{E}\left(\sum_{t=1}^{T}\frac{\partial\boldsymbol{s}^l(t)}{\partial\boldsymbol{m}^l(t)}\right)\left(\frac{1}{T}\sum_{t=1}^{T}\frac{\partial\boldsymbol{m}^l(t)}{\partial\boldsymbol{s}^{l-1}(t)}\right) \\
=& \frac{\partial\mathcal{L}}{\partial\boldsymbol{r}^l(T)}\left(\frac{\partial\boldsymbol{r}^l(T)}{\partial\boldsymbol{W}^l\boldsymbol{r}^{l-1}(T)}\right)_{\text{rate}}\frac{\partial\boldsymbol{W}^l\boldsymbol{r}^{l-1}(T)}{\partial\boldsymbol{r}^{l-1}(T)} \\
=& \left(\frac{\partial\mathcal{L}}{\partial\boldsymbol{r}^{l-1}(T)}\right)_{\text{rate}}.
\end{aligned}
\tag{A5}
$$

$\square$

Table A1: ASR about STBP/STBP+optimal $\gamma$/Ours for CIFAR-10/VGG-11.

| $\lambda$ | FGSM, WBA | FGSM, BBA | PGD, WBA | PGD, BBA |
|---|---|---|---|---|
| 0.5 | 91.71/91.97/**96.28** | 77.77/77.90/**82.68** | 99.95/99.95/**100.00** | 95.99/96.03/**96.74** |
| 1.0 | 86.28/90.40/**96.22** | 68.03/72.78/**80.34** | 99.70/99.81/**99.97** | 90.91/94.06/**96.22** |

Table A2: High-intensity attack about STBP/RGA/Ours for robust SNNs, $\lambda$=0.9, VGG-11.

| CIFAR-10, WBA | CIFAR-10, BBA | CIFAR-100, WBA | CIFAR-100, BBA |
|---|---|---|---|
| 84.71/81.01/**88.12** | 56.82/54.79/**58.16** | 90.61/88.26/**91.65** | 65.94/65.69/**68.00** |

## A.2 FURTHER EXPERIMENTS & PARAMETER CONFIGURATION

### A.2.1 STBP ATTACK WITH OPTIMAL $\gamma$

In previous works (Sharmin et al., 2019; Bu et al., 2023), the vanilla STBP attack directly adopted a back-propagation mode consistent with the STBP learning algorithm to obtain the disturbance gradients. In this work, we consider a more threatening STBP attack that enhances the attack success rate by adjusting the value of $\gamma$. Tab. A1 presents the attack results of our proposed method and two types of STBP attacks. It is evident that the HART attack framework consistently outperforms both types of STBP attacks in all cases, which further verifies the effectiveness of our gradient pruning and merging techniques.

### A.2.2 PGD ATTACK WITH HIGHER INTENSITY

We randomly restart the 10-step PGD attack 5 times and calculate the cumulative attack success rate during this procedure. We set the robust SNNs used in Tab. 2 as our attack targets. The results of this experiment are presented in Tab. A2. It is evident that the HART attack continues to maintain a significant advantage in terms of attack success rate.

### A.2.3 MEDIAN FILTER

We also consider the impact of filtering on our proposed attack method from the perspective of traditional image processing. During the training and testing process, we chose the median filtered image as the input data. As shown in Tab. A3, compared to STBP and RGA, HART exhibits significant advantages at different time-steps. For example, when adopting 8 time-steps, our method has a performance advantage of at least 3.47% on FGSM attack.

### A.2.4 POISSON ENCODING

Poisson encoding is a widely used neuromorphic encoding scheme for SNNs, which simulates spike sequences by binarizing each pixel point. The pixel value is proportional to the corresponding spike firing rate. We have verified that HART can still achieve superior attack performance under the Poisson encoding environment. As shown in Tab. A4, our method consistently achieves optimal results in various experimental cases.

### A.2.5 TRAINING AND INFERENCE PARAMETER SETTING

In Tabs. 2-3, we set the perturbation strength $\epsilon = 8/255$ for all experimental cases about vanilla SNN models. The step size $\alpha$ and the number of iterations for the PGD attack is set to $2/255$ and 5, respectively. For robust SNN models, we adopt adversarial STBP training for 4 time-steps and the iteration number of PGD is set to 10. Regarding neuromorphic data, following the approach described in (Bu et al., 2023), the perturbation obtained through gradient calculation is directly allocated to each preprocessed data frame. Additionally, considering that neuromorphic data may contain abundant temporal features, we also attempt to use a combination of the gradient obtained from STBP and HART as the final gradient for the attack. We compared the optimal attack performance between the HART gradient and the mixed gradient for neuromorphic datasets.

Table A3: ASR about STBP/RGA/Ours with median filter under WBA for CIFAR-10/VGG-11.

| Time-steps | FGSM, $\lambda = 0.9$ | FGSM, $\lambda = 1.0$ | PGD, $\lambda = 0.9$ | PGD, $\lambda = 1.0$ |
|---|---|---|---|---|
| 4 | 73.88/79.45/**80.80** | 75.45/81.02/**83.00** | 99.06/99.38/**99.41** | 97.78/98.31/**99.02** |
| 8 | 78.69/82.49/**85.96** | 79.31/83.48/**87.05** | 99.56/99.68/**99.83** | 99.36/99.69/**99.81** |
| 16 | 79.47/83.46/**87.64** | 80.04/85.44/**88.81** | 99.68/99.80/**99.91** | 99.61/99.77/**99.89** |

Table A4: ASR about STBP/RGA/Ours with Poisson encoding for CIFAR-10/VGG-11.

| $\lambda$ | Time-steps | FGSM, WBA | FGSM, BBA | PGD, WBA | PGD, BBA |
|---|---|---|---|---|---|
| 0.5 | 16 | 71.95/64.78/**77.47** | 46.46/42.26/**50.98** | 81.33/72.11/**85.27** | 50.01/44.59/**54.45** |
| | 32 | 72.73/61.73/**76.45** | 52.47/44.67/**56.20** | 81.01/69.70/**83.88** | 57.31/47.73/**60.99** |
| 1.0 | 16 | 77.13/75.68/**79.10** | 40.77/44.08/**47.63** | **87.08**/86.48/87.02 | 43.04/47.21/**50.68** |
| | 32 | 77.15/75.93/**80.40** | 46.26/50.66/**55.12** | 87.39/86.91/**88.37** | 50.05/55.28/**59.32** |

**Optimizer & Scheduler.** During the training procedure, we chose the Stochastic Gradient Descent Optimizer (SGD) in conjunction with the Cosine Annealing scheduler (Loshchilov & Hutter, 2016) to the STBP method for 200 epochs. The initial learning rate, momentum, and weight decay are set to 0.1, 0.9, and $5 \times 10^{-4}$, respectively.

**Attack Scenarios.** For white-box attacks, we directly calculate the gradient based on the back-propagation chain of different methods and add it as a perturbation to the input data. For black-box attacks, similar to (Bu et al., 2023), our gradient is calculated using the black-box model, which is a similar model trained through different random seeds compared to the attack target model.

**Mixed Gradient.** For neuromorphic datasets, to further enhance the temporal properties of our attack scheme, we attempt to combine the gradient obtained from HART and STBP. The mixed gradient can be described as follows:

$$(\nabla_{\boldsymbol{x}}\mathcal{L}(f(\boldsymbol{x}, \boldsymbol{W}), y))_{\text{MIX}} = \beta \left(\nabla_{\boldsymbol{x}}\mathcal{L}(f(\boldsymbol{x}, \boldsymbol{W}), y)\right)_{\text{HART}} + (1 - \beta) \left(\nabla_{\boldsymbol{x}}\mathcal{L}(f(\boldsymbol{x}, \boldsymbol{W}), y)\right)_{\text{STBP}}.$$
(A6)

In our experiments, we choose a mixture of equal proportions ($\beta = 0.5$) to validate the attack performance.

## A.3 DETAILED ALGORITHM FOR THE HART ATTACK FRAMEWORK

**Algorithm 1** Hybrid Adversarial Attack by Rate and Temporal Information (HART).

**Require:** Input image $\boldsymbol{x}$; Weight matrix and spiking neuron layer $\boldsymbol{W}^l, \textbf{Neuron}^l, \forall l \in [1, L]$; Loss function $\mathcal{L}$; Total simulation time $T$.
**Ensure:** Adversarial example image $\hat{\boldsymbol{x}}$.
 1: # Forward propagation
 2: **for** $l = 1$ to $L$ **do**
 3:     **for** $t = 1$ to $T$ **do**
 4:         Update the state of **Neuron**$^l$ based on Eq.(1)
 5:         Determine whether to fire spikes based on Eq.(4)
 6:         Reset the state of **Neuron**$^l$ based on Eq.(2)
 7:         **if** Use the pre-calculation property **then**
 8:             Calculate $\sum_{t=1}^{T} \frac{\partial \boldsymbol{s}^l(t)}{\partial \boldsymbol{m}^l(t)}$ by Eq.(15).
 9:         **end if**
10:     **end for**
11: **end for**
12: # Back-propagation
13: **for** $l = L$ to $1$ **do**
14:     **if** Not use the pre-calculation property **then**
15:         **for** $t = 1$ to $T$ **do**
16:             Calculate the surrogate gradient $\frac{\partial \boldsymbol{s}^l(t)}{\partial \boldsymbol{m}^l(t)}$ at each time-step by Eq.(15)
17:         **end for**

18:   Calculate the gradient merging term $\left[\frac{\partial \mathcal{L}}{\partial \boldsymbol{m}}\right]$ by Eq.(14)
19:   **for** $t = 1$ to $T$ **do**
20:       Calculate $\frac{\partial \mathcal{L}}{\partial \boldsymbol{s}^{l-1}(t)} = \left[\frac{\partial \mathcal{L}}{\partial \boldsymbol{m}}\right] \frac{\partial \boldsymbol{m}^l(t)}{\partial \boldsymbol{s}^{l-1}(t)}$ at each time-step by Eq.(13)
21:   **end for**
22:   **else**
23:       As $\frac{\partial \mathcal{L}}{\partial \boldsymbol{s}^l(1)} = ... = \frac{\partial \mathcal{L}}{\partial \boldsymbol{s}^l(T)}$, here we use $\frac{\partial \mathcal{L}}{\partial \boldsymbol{s}^l}$ to denote them
24:       Calculate the gradient merging term $\left[\frac{\partial \mathcal{L}}{\partial \boldsymbol{m}}\right] = \frac{\partial \mathcal{L}}{\partial \boldsymbol{s}^l} \sum_{t=1}^{T} \frac{\partial \boldsymbol{s}^l(t)}{\partial \boldsymbol{m}^l(t)}$
25:       Calculate $\frac{\partial \mathcal{L}}{\partial \boldsymbol{s}^{l-1}} = \boldsymbol{W}^{l^\top} \left[\frac{\partial \mathcal{L}}{\partial \boldsymbol{m}}\right]$
26:   **end if**
27: **end for**
28: Calculate the perturbation gradient $\nabla_{\boldsymbol{x}} \mathcal{L}$ about the input image $\boldsymbol{x}$
29: For FGSM attack, calculate $\hat{\boldsymbol{x}} = \boldsymbol{x} + \epsilon \text{sign}(\nabla_{\boldsymbol{x}} \mathcal{L})$ by Eq.(6)
30: For PGD attack, calculate $\hat{\boldsymbol{x}}^k = \prod_{\boldsymbol{x},\epsilon} \{\hat{\boldsymbol{x}}^{k-1} + \alpha \text{sign}(\nabla_{\boldsymbol{x}} \mathcal{L})\}$ by Eq.(7)
31: **return** Adversarial example image $\hat{\boldsymbol{x}}$

## A.4   THE COMPLEXITY OF GRADIENT COMPUTATION IN HART

As discussed in the **Pre-calculation Property** section, we employ a pre-calculation technique for the spiking neuron layers in the HART framework. This involves calculating the sum of surrogate gradients across multiple time-steps. By doing so, we can compute the gradient of HART in a single step during the back-propagation process, similar to ANNs. Consequently, the time complexity of pre-calculation for spiking neurons is $O(T)$, while the gradient computation complexity for each synaptic layer remains at $O(1)$. In the case of deeper SNNs with more synaptic layers, we believe that this property will lead to a more significant reduction in computational cost compared to the vanilla STBP attack.

## A.5   SPECIFIC EXPLANATION ABOUT FIGURE 1

Fig. 1 depicts a scene where two presynaptic neurons are connected to one postsynaptic neuron. In all three cases, we set $\boldsymbol{W}^l = [3\theta^l/2, -\theta^l/2]$ and $\boldsymbol{r}^{l-1}(T) = [1/2, 1/2]^T$. In Case 1, positive and negative currents are alternately inputted, resulting in the postsynaptic neuron emitting spikes at moments 1, 3, and 5. In Case 2, a continuous positive current is inputted during the first three moments, leading to an increased number of spikes in the postsynaptic layer. In Case 3, negative currents are inputted during the first three time-steps, resulting in fewer spikes in the postsynaptic neuron.

## A.6   DETAILED DERIVATION FOR EQUATION 8

By combining Eq.1 and Eq.2, we have:

$$\boldsymbol{v}^l(t) - \lambda^l \boldsymbol{v}^l(t-1) = \boldsymbol{W}^l \boldsymbol{s}^{l-1}(t) - \boldsymbol{\eta}^l(t) \boldsymbol{s}^l(t).$$

As we adopt the soft-reset mechanism and set $\theta^l = 1$, we further have:

$$\boldsymbol{v}^l(t) - \lambda^l \boldsymbol{v}^l(t-1) = \boldsymbol{W}^l \boldsymbol{s}^{l-1}(t) - \boldsymbol{s}^l(t).$$

By summing the above equation from $t = 1$ to $t = T$, and dividing both sides by $T$, we have:

$$\frac{\boldsymbol{v}^l(T) + \sum_{t=1}^{T-1}(1 - \lambda^l)\boldsymbol{v}^l(t) - \lambda^l \boldsymbol{v}^l(0)}{T} = \boldsymbol{W}^l \frac{\sum_{t=1}^{T} \boldsymbol{s}^{l-1}(t)}{T} - \frac{\sum_{t=1}^{T} \boldsymbol{s}^l(t)}{T}.$$

As we set $\boldsymbol{v}^l(0) = 0, \boldsymbol{r}^l(T) = \sum_{t=1}^{T} \boldsymbol{s}^l(t)/T$, finally we get:

$$\boldsymbol{r}^l(T) = \boldsymbol{W}^l \boldsymbol{r}^{l-1}(T) - \frac{\boldsymbol{v}^l(T) + \sum_{t=1}^{T-1}(1 - \lambda^l)\boldsymbol{v}^l(t)}{T}.$$

