# OpenReview forum: "Threaten Spiking Neural Networks through Combining Rate and Temporal Information"
_ICLR.cc/2024/Conference — ICLR 2024 poster_

### Official Review · Reviewer_4VCV · 2023-10-24

**Soundness:** 2 fair
**Presentation:** 2 fair
**Contribution:** 3 good
**Rating:** 6
**Confidence:** 4

**Summary:**

The paper presents HART, an adversarial attack for SNNs. An analysis of rate and temporal information in SNNs identifies the importance of rate information in SNNs. The proposed attack combines rate and temporal information. The results show better attack success rates than related works on CIFAR-10 and CIFAR-10-DVS datasets.

**Strengths:**

1. The tackled problem is relevant to the community.

2. The proposed attack is novel.

3. The results outperform related works.

**Weaknesses:**

Some aspects are not completely clear. Please see the questions below.

**Questions:**

1. In Table 1, please clarify what results are relative to CBA and what results are relative to the proposed attack.

2. In Section 4.2: “Therefore, we attempt to measure the retention degree of temporal information in SNNs through the following equation.” Please provide more details of the intuitions and design decisions made to develop Eq.12.

3. Please describe the proposed attack through a detailed algorithm that collects all the operations and equations involved. The current description with several equations may be unclear.

4. Section 5 contains the results when varying the parameters of the attack. However, there is very little discussion on the results and reasons why certain values of the parameters achieve higher attack success rates than others. Please discuss it in a more comprehensive manner.

---

> ### Author Response · Authors · 2023-11-19
> **To Reviewer 4VCV (Part I)**
>
> ## To Reviewer 4VCV
> Thanks for your invaluable and constructive feedback! We are encouraged that you find our paper novel and relevant to the community, as well as outperforming related works. We would like to address your concerns and your questions in the following.
>
>
> **Q1: In Table 1, please clarify what results are relative to CBA and what results are relative to the proposed attack.**
>
> **A1:** Thanks for pointing it out! In Table 1, the regular font on the left side of the table represents CBA (the baseline method), while the bold font on the right side represents our proposed attack method. We have made a clarification in our newly submitted version.
>
>
> **Q2: Please provide more details of the intuitions and design decisions made to develop Eq.12.**
>
> **A2:** From Figure 1, it is evident that the same average input current ($W^lr^{l-1}(T)$) can lead to different spike firing rates ($r^l(T)$) due to the non-uniform distribution of input current across multiple consecutive time-steps. Previous studies on ANN-SNN conversion [1, 2] have found that assuming a uniform input current, we have the relationship $r^l(T) = \frac{1}{T}\text{clip}\left( \left\lfloor \frac{TW^lr^{l-1}(T)+v^l(0)}{\theta^l} \right\rfloor, 0, T \right)$. This implies that the average firing rate of adjacent layers follows an approximate linear transformation, and there exists a one-to-one mapping between $r^l(T)$ and $W^lr^{l-1}(T)$. Although the equation $r^l(T) = \frac{1}{T}\text{clip}\left( \left\lfloor \frac{TW^lr^{l-1}(T)+v^l(0)}{\theta^l} \right\rfloor, 0, T \right)$ is based on the IF model, it holds true that the one-to-one mapping relationship still applies to general LIF models. In this scenario, the SNN is predominantly influenced by rate information, and the computational relationship between two adjacent SNN layers is entirely equivalent to that of an ANN. Moreover, in this situation, for any $W^lr^{l-1}(T)$, the variance of $\frac{r^l(T)}{W^lr^{l-1}(T)}$ is zero, that is, $\mathbf{Var}\left( \frac{r^l(T)}{W^lr^{l-1}(T)} \right)=0$.
>
> For SNNs with more temporal information, the same average input current ($W^lr^{l-1}(T)$) can lead to different spike firing rates ($r^l(T)$), and the variance of $\frac{r^l(T)}{W^lr^{l-1}(T)}$ is not zero. Thus, it is reasonable to consider to use the variance of $\frac{r^l(T)}{W^lr^{l-1}(T)}$ to measure the temporal information and we propose Eq. 12 to measures the richness of temporal information according to the expectation of the variance. As $\chi^l$ increases, the variations in firing more or fewer spikes due to different spike arrival sequences in Figure 1 become more pronounced. This signifies a stronger influence of temporal information within the SNN.
>
>
> **Q3: Please describe the proposed attack through a detailed algorithm that collects all the operations and equations involved.**
>
> **A3:** Thanks for your constructive suggestion! We have made a detailed algorithm to describe our HART attack framework step by step. Please refer to Section A.3 in the Appendix of our newly submitted version.

---

> > ### Author Response · Authors · 2023-11-19
> > **To Reviewer 4VCV (Part II)**
> >
> > **Q4: There is very little discussion on the results and reasons why certain values of the parameters achieve higher attack success rates than others. Please discuss it in a more comprehensive manner.**
> >
> > **A4:** Thanks for your invaluable question! Regarding static datasets, the attack success rate (ASR) of HART generally remains consistent across different values of the parameter $\lambda$. We attribute this performance to the dynamic adjustment capability of HART in terms of temporal attributes. By selecting suitable values of $\gamma$ for specific scenarios ($\lambda$), as discussed in Section 4.3 ("Empirical Principles for Selecting $\gamma$"), we can effectively adapt the framework. It is reasonable to observe a decrease in ASR for all methods when dealing with smaller perturbation degrees (as shown in Table 5) or robust target models trained using adversarial samples (as seen in Tables 2 and 3, marked with "$*$").
> >
> > For neuromorphic datasets, it is important to consider that the input data at each time-step (frame) represents the integration result of the event camera over a certain period. Consistent with the experimental settings in [3], we directly allocate the calculated HART gradient to each frame of the data. However, due to the generally lower ASR results obtained by all methods, further research is required to determine the appropriateness of the gradient perturbation method. For different time-steps, one can find that when the number of time-steps used in the attack is generally equal to or greater than the training steps of the target model (e.g., $\text{Time-steps} \geq 8$ in Section 5, Table 4), the attack performance of the method tends to stabilize. This is because too few number of time-steps are inadequate for accurately extracting the rate and temporal information of SNNs.
> >
> > [1] Yuhang Li, Shikuang Deng, Xin Dong, Ruihao Gong, and Shi Gu. A free lunch from ANN: Towards efficient, accurate spiking neural networks calibration. In International Conference on Machine Learning, pp. 6316–6325, 2021.
> >
> > [2] Tong Bu, Wei Fang, Jianhao Ding, PengLin Dai, Zhaofei Yu, and Tiejun Huang. Optimal ANN-SNN conversion for high-accuracy and ultra-low-latency spiking neural networks. In International Conference on Learning Representations, 2022.
> >
> > [3] Tong Bu, Jianhao Ding, Zecheng Hao, and Zhaofei Yu. Rate gradient approximation attack threats deep spiking neural networks. In IEEE/CVF Conference on Computer Vision and Pattern Recognition, 2023.

---

> > > ### Comment · Reviewer_4VCV · 2023-11-23
> > > **Response to Authors' Rebuttal**
> > >
> > > Thank you for your responses. Considering together the other reviews and responses, my score is confirmed.

---

### Official Review · Reviewer_w1gJ · 2023-10-29

**Soundness:** 3 good
**Presentation:** 3 good
**Contribution:** 3 good
**Rating:** 6
**Confidence:** 5

**Summary:**

This paper analyzes the temporal information in SNNs and establishes its relationship with the membrane decay constant and the number of time-steps. Then the authors propose a novel approach to combine rate and temproal information in SNNs, leveraging it to generate effective gradients for attacking SNNs. The experimental results demonstrate the proposed method achieve the SOTA attack success rate.

**Strengths:**

1. This paper is well-written.
2. The idea of combining rate and temporal information in SNNs is highly noteworthy. The temporal information in SNNs has not been fully utilized in the current model.
3. The work is solid. The authors provide a rigorous theoretical analysis of the retention degree of temporal information in SNNs and showcase the influencing factors involved.

**Weaknesses:**

1. My main concern is about equation 8. I find it hard to derive equation 8 by combining equations 1 and 2. I am wondering about the absence of the threshold in equation 8. The authors should clarify why the threshold is not included in equation 8.
2. The authors primarily focus on demonstrating how the combination of rate temporal information can enhance the attack of SNNs. How about defense? Can the proposed method also be used to improve the robustness of SNNs.

**Questions:**

Refer to the weaknesses above.
Please provide a more comprehensive explanation of Figure 2(c).

---

> ### Author Response · Authors · 2023-11-19
> **To Reviewer w1gJ**
>
> ## To Reviewer w1gJ
> Thanks for your insightful and constructive feedback! We are encouraged that you find our paper novel, solid, well-written and highly noteworthy, as well as providing a rigorous theoretical analysis. We would like to address your concerns and your questions in the following.
>
> **Q1: I find it hard to derive equation 8 by combining equations 1 and 2. I am wondering about the absence of the threshold in equation 8. The authors should clarify why the threshold is not included in equation 8.**
>
> **A1:** Thanks for pointing it out. To simplify the subsequent mathematical analysis, we actually set $\theta^l=1$ here. We have clarified in page 3 and added detailed derivation in the Appendix, Section A.6.
>
> **Q2: How about defense? Can the proposed method also be used to improve the robustness of SNNs.**
>
> **A2:** Thanks for your constructive comment! We have explored the use of images perturbed by the HART gradient as adversarial samples to enhance the robustness of SNNs through adversarial training. Our experiments encompass both vanilla SNNs and robust SNNs obtained via HART adversarial training, evaluated under FGSM and PGD attack scenarios with a white-box setting. We employ the attack success rate as our metric and set $\lambda$ to 0.9.
> As depicted in Table R1, our results demonstrate a significant improvement in the defense capability of robust SNNs across various attack scenarios, including BPTR, STBP, and RGA attacks. Notably, during our adversarial training, we did not introduce adversarial samples specifically tailored to these three mentioned attack algorithms. This finding indicates that HART not only enhances the defense capability of SNNs but also exhibits superior generalization ability, making it applicable for defending against a wide range of attack algorithms.
>
>
> **Table R1: Comparison between the vanilla and robust models on the CIFAR-100 dataset.**
> | Dataset | model | Architecture | attack | BPTR | BPTT | RGA |
> | ------- | --- | ------------ | ------ | ---- | ---- | --- |
> | CIFAR-100 | vanilla | VGG-11 | FGSM |92.47|92.88|94.72|
> | CIFAR-100 | robust | VGG-11 | FGSM |49.09|59.31|57.45|
> | CIFAR-100 | vanilla | VGG-11 | PGD |99.59|99.86|99.92|
> | CIFAR-100 | robust | VGG-11 | PGD* |59.51|72.46|69.36|
>
> $*$ denotes a stronger PGD attack ($\alpha=2.55,\text{steps}=7$).
>
> **Q3: Please provide a more comprehensive explanation of Fig.2$\text{(c)}$.**
>
> **A3:** Thank you for your insightful comment! We have made improvements to Fig. 2$\text{(c)}$  in our newly submitted version. In the updated figure, we illustrate the impact of different scales of $\gamma$ on the gradient discrepancy. We specifically select three spiking neurons with distinct $m^l(t)$ values at the $t$-th time-step. It can be observed that as $\gamma$ decreases, the discrepancy of surrogate gradients among the spiking neurons increases. Consequently, HART tends to focus on a subset of spiking neurons that fall within a specific membrane potential range at each time-step. This leads to the gradient with enhanced temporal attributes. Conversely, as $\gamma$ increases, the gradient discrepancy among the three spiking neurons diminishes, and HART exhibits a more pronounced rate attribute.

---

### Official Review · Reviewer_RoZ6 · 2023-10-30

**Soundness:** 3 good
**Presentation:** 3 good
**Contribution:** 3 good
**Rating:** 8
**Confidence:** 4

**Summary:**

This paper proposes an adversarial attack method for SNN, which combines both rate and temporal information. Based on detailed analysis and experiments, this work demonstrates the efficiency of the proposed method.

**Strengths:**

1.	It is interesting to consider both rate and temporal characters in SNN when considering adversarial attack.
2.	Author provide detailed experiment to demonstrate the effectiveness of the proposed framework.

**Weaknesses:**

1.	The theory is hard to follow, it is better to provide an illustration of how to utilize temporal information during BP (as Fig.1)
2.	Does the pruning has negative effect on the gradient computation? I think it is also interesting to discuss whether the proposed gradient computation can be directly applied to SNN training.
3.	It is better to discuss the complexity of gradient computation in HART.
4.	Fig.2(c) is hard to understand. In Fig.2(b), it is not clear the meaning of solid arrows.

**Questions:**

Overall, I think this work provide an interesting attack method, and the experiments are well presented. The theory is not easy to follow, please see my comments for detail.

---

> ### Author Response · Authors · 2023-11-19
> **To Reviewer RoZ6**
>
> ## To Reviewer RoZ6
> Thanks for your constructive and thoughtful comments! We are encouraged that you find our paper interesting and effective, as well as providing detailed experiments. We would like to address your concerns and your questions in the following.
>
> **Q1: It is better to provide an illustration of how to utilize temporal information during BP (as Fig.1).**
>
> **A1:** Thanks for your advice. We have improved Figure 2 and the related content to better illustrate the details of utilizing temporal information during BP in our newly submitted version.
>
> **Q2: Does the pruning has negative effect on the gradient computation? Whether the proposed gradient computation can be directly applied to SNN training?**
>
> **A2:** Thanks for your insightful comment! We would like to clarify that in this work, the gradient pruning and merging techniques are combined within the HART framework to enhance the extraction of rate and temporal information. This combination has a positive impact on improving attack success rates. In Table A1 (found in the Appendix), we present a series of ablation experiments that compare "STBP+optimal $\gamma$" with HART. The key distinction between the two lies in the inclusion or exclusion of pruning and merging. The experimental results demonstrate that pruning and merging indeed contribute to improved attack effectiveness. Thanks for your suggestion! We agree that the potential application of these gradient techniques to SNN training is a fascinating direction for further research. We have included this point in the conclusion section.
>
>
> **Q3: It is better to discuss the complexity of gradient computation in HART.**
>
> **A3:** Thanks for pointing it out. As discussed in the "Pre-calculation Property" section, we employ a pre-calculation technique for the spiking neuron layers in the HART framework. This involves calculating the sum of surrogate gradients across multiple time-steps. By doing so, we can compute the gradient of HART in a single step during the back-propagation process, similar to ANNs. Consequently, the time complexity of pre-calculation for spiking neurons is $O(T)$, while the gradient computation complexity for each synaptic layer remains at $O(1)$. In the case of deeper SNNs with more synaptic layers, we believe that this property will lead to a more significant reduction in computational cost compared to the vanilla STBP attack. We have also added the discussion in the Appendix, Section A.4.
>
>
> **Q4: In Fig.2(b), it is not clear the meaning of solid arrows. Fig.2$\text{(c)}$ is hard to understand.**
>
> **A4:** Thanks for pointing it out. We have made improvements to Fig. 2(b) and Fig.2$\text{(c)}$ in the revised version. In the updated figures, the solid lines representing gradient pruning and merging are connected to Fig. 2(a), indicating that the design of these techniques ensures the establishment of the pre-calculation property in HART during forward propagation. The solid line related to the adjustable surrogate gradient is connected to Fig.2$\text{(c)}$. In Fig.2$\text{(c)}$, we illustrate the impact of different scales of $\gamma$ on the gradient discrepancy. We have selected three spiking neurons with varied $m^l(t)$ values at the $t$-th time-step. It can be observed that as $\gamma$ decreases, the discrepancy of surrogate gradients among spiking neurons increases. This causes HART to focus on a subset of spiking neurons within a specific membrane potential range at each time-step. Consequently, this enables us to observe a gradient with more pronounced temporal attributes.

---

> > ### Comment · Reviewer_RoZ6 · 2023-11-23
> > **Response to authors**
> >
> > Thanks for the detailed response to address my concerns. The design framework is much clearer for me now. The exploration of temporal aspects and incorporating ratings in the context of SNN security is an intriguing research direction. I appreciate this innovative approach and would like to increase my score accordingly.

---

### Official Review · Reviewer_hCfa · 2023-10-31

**Soundness:** 3 good
**Presentation:** 4 excellent
**Contribution:** 4 excellent
**Rating:** 8
**Confidence:** 4

**Summary:**

This paper proposed a hybrid adversarial attack framework, named HART, based on both rate and temporal information. The proposed method offers the flexibility to dynamically adjust the proportion between rate and temporal attributes according to the defined variable known as the retention degree of temporal information. Experiments indicated that HART can significantly improve the attack success rate of SNNs in various attack scenarios.

**Strengths:**

1.The analytical perspective of this paper is novel. The author starts with the propagation mode between SNN layers and summarizes the two types of information: rate and temporal information. Then they explored how to obtain more precise attack gradients by analyzing the optimal integration methods for both types of information.

2.This paper takes an important initial stride in showcasing the potential of combining rate and temporal information to enhance SNN attacks.

3.The theoretical derivation and mathematical analysis encompassed in this paper are solid.

4. The results of extensive experiments effectively demonstrate that HART exhibits remarkable performance improvements under different hyperparameter configurations when compared to previous SOTA works.

**Weaknesses:**

1. The authors should provide a more detailed analysis on the specific role of gradient pruning and merging within the proposed framework.

2. The figures and legends of the paper can be further improved.

**Questions:**

1. In Figure 1, it is unclear why the purple curve appears to be identical to the curve representing case 1.

2.  I kindly request the authors to assess which specific information (rate or temporal) from the SNN was leveraged by several previous attack algorithms, namely CBA, STBP, BPTR, and RGA, in their experimental evaluations. It may shed light on the attack and defense exploration.

3. In Table 1, the authors juxtapose their rate-based gradient attack method with CBA. Given that CBA also employs rate information for its attack strategy, could the authors elucidate the factors that make the proposed rate gradient method superior?

---

> ### Author Response · Authors · 2023-11-19
> **To Reviewer hCfa**
>
> ## To Reviewer hCfa
> Thanks for your insightful and invaluable comments! We are encouraged that you find our paper novel and solid, as well as taking an important initial stride, having theoretical derivations and exhibiting remarkable performance improvements. We would like to address your concerns and your questions in the following.
>
> **Q1: The authors should provide a more detailed analysis on the specific role of gradient pruning and merging.**
>
> **A1:** Thanks for your constructive comment! Gradient pruning and merging have two primary functions. Firstly, they facilitate the comprehensive utilization of both rate and temporal information. Pruning prevents gradient interference between different time-steps, while merging performs overall statistics for surrogate gradients, specifically the membrane potential information, across multiple time-steps. The synergistic use of pruning and merging promotes the integration of rate and temporal information, thereby guaranteeing the establishment of Theorem 2. Secondly, gradient pruning and merging ensure the desirable pre-calculation property of the HART framework, which requires only a single-time gradient calculation for each synaptic layer, effectively reducing the computational complexity of the algorithm.
>
>
> **Q2: The figures and legends of the paper can be further improved.**
>
> **A2:** Thanks for your advice! We have made modifications for the figures and legends in this paper, and submitted a new version.
>
> **Q3: In Fig.1, it is unclear why the purple curve appears to be identical to the curve representing case 1.**
>
> **A3:** In fact, there is no correlation between these two curves. We have revised the colors of Fig.1 in our newly submitted version.
>
> **Q4: I kindly request the authors to assess which specific information (rate or temporal) was leveraged by CBA, STBP, BPTR, and RGA.**
>
> **A4:**  CBA directly transfers the gradients obtained from pre-trained ANNs to SNNs by leveraging the rate information, assuming a uniform input current. STBP utilizes gradients calculated through spatial-temporal back-propagation across multiple consecutive time steps, primarily relying on temporal information. BTPR and RGA employ the vanilla forward propagation of spiking neurons while incorporating rate information during gradient calculation. BPTR masks the gradient based on whether a neuron has emitted spikes, while RGA replaces the surrogate gradient function used in STBP with the activation gradient obtained from the Rate-Input Curve.
>
> **Q5: Given that CBA also employs rate information for its attack strategy, could the authors elucidate the factors that make the proposed rate gradient method superior?**
>
> **A5:** Thanks for pointing it out. In terms of forward propagation, CBA relies on the assumption of uniform input current to determine the output in each layer, whereas our approach utilizes the actual output of spiking neurons across multiple time steps. When it comes to gradient calculation, CBA typically employs a conventional activation function like ClipReLU, whereas we derive our activation gradient by averaging the statistical information of spike firing on a layer-by-layer basis. We believe that these two aspects have enabled our method to achieve more precise gradient estimations.

---

> > ### Comment · Reviewer_hCfa · 2023-11-22
> > **Responses**
> >
> > Thanks for the response.
> > The authors have well addressed my concerns， I recommend acceptance.

---

### Meta-Review · Area_Chair_yn19 · 2023-12-04

**Metareview:**

This paper examines adversarial attacks on spiking neural networks (SNNs). The authors develop an adversarial attack technique that uses both spike rate and spike-timing information, called HART, which they show works better than previous white-box attack methods for SNNs.

The reviewers were fairly positive about this paper, though there were consistent concerns about clarity in parts. The authors rebuttals helped to aleviate most of the concerns raised though, and the final scores were all above threshold. Therefore, a decision of 'accept' was reached.

**Justification For Why Not Higher Score:**

Though the reviewers were clear that this paper is worth accepting, it is a fairly niche paper - adversarial attacks for SNNs are not a big topic for ICLR's audience, I would say. As well, the paper is not super well-written, even after some edits (it's good enough to accept, but not good enough to spotlight, IMO).

**Justification For Why Not Lower Score:**

The reviewers were all clear that this should be accepted.

---

### Decision · Program_Chairs · 2024-01-16

Accept (poster)